# Improving Interaction Comfort in Authoring Task in AR-HRI through Dynamic Dual-Layer Interaction Adjustment

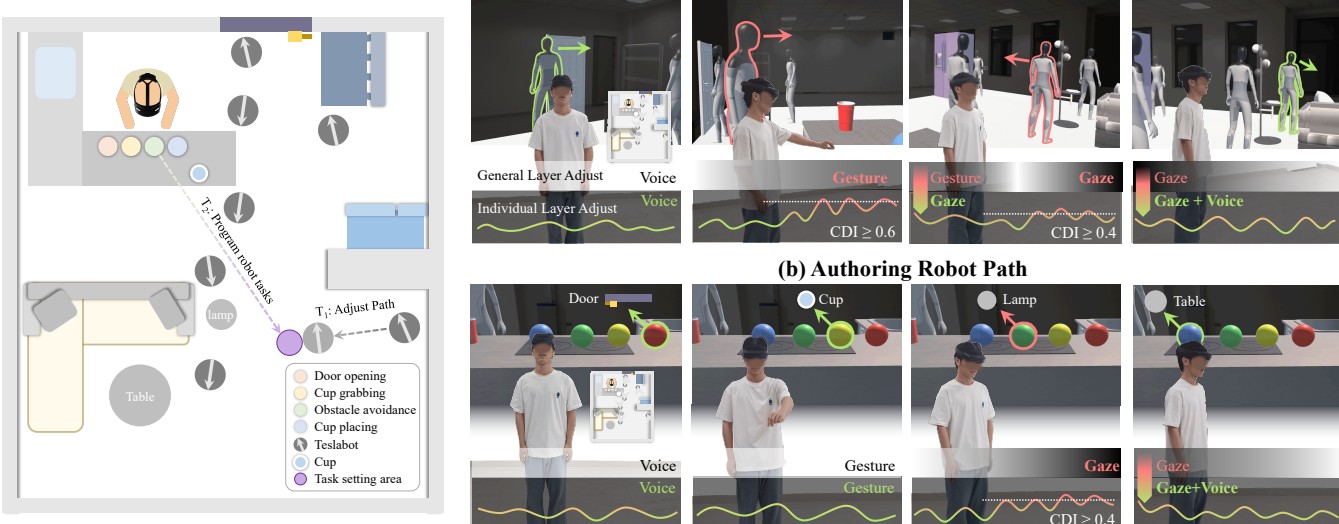

**(a) Welcoming Guests Task**

**(b) Authoring Robot Path**

**(c) Programming Robot Task**

**Figure 1: In the authoring tasks for "Robot Welcoming Guests", participants adjusted the robot's initial path and programmed its behaviors for obstacle avoidance, door opening, and handling a cup. The interaction method is dynamic, adapting through general and individual layer modeling. The general model segments space for ergonomic interactions, while the individual layer model uses physiological signals to predict discomfort and adjust interactions accordingly. Adjustments are triggered when the Continuous Discomfort Index (CDI) exceeds a certain threshold.**

## ABSTRACT

Previous research has demonstrated the potential of Augmented Reality in enhancing psychological comfort in Human-Robot Interaction (AR-HRI) through shared robot intent, enhanced visual feedback, and increased expressiveness and creativity in interaction methods. However, the challenge of selecting interaction methods that enhance physical comfort in varying scenarios remains. This study purposes a dynamic dual-layer interaction adjustment mechanism to improve user comfort and interaction efficiency. The mechanism comprises two models: an general layer model, grounded in ergonomics principles, identifies appropriate areas for various interaction methods; a individual layer model predicts user discomfort levels using physiological signals. Interaction methods are dynamically adjusted based on continuous discomfort level changes, enabling the system to adapt to individual differences and

dynamic changes, thereby reducing misjudgments and enhancing comfort management. The mechanism's success in authoring tasks validates its effectiveness, significantly advancing AR-HRI and fostering more comfortable and enhancing efficient human-centered interactions.

## CCS CONCEPTS

• **Human-centered computing → Interaction paradigms**; **Interaction design**.

## KEYWORDS

Interaction Comfort, Augmented Reality; Physiological Computing; Human-Robot Interaction;

**Unpublished working draft. Not for distribution.**

## 1 INTRODUCTION

Comfort is considered as an important indicator of interaction quality in human-robot interaction (HRI) [47]. Recent studies on AR-enhanced Human-Robot Interaction (AR-HRI) [43] have shown that AR can enhance human comfort in HRI by sharing robot's intent [52] and plan [3], enhancing visual feedback [24], and increasing the expressiveness and creativity of interaction methods [13]. Comfort is mainly divided into psychological comfort and physical comfort [44]. Although AR-HRI has made some progress

in improving psychological comfort, how to choose the appropriate interaction method for the current scenario to improve physical comfort is still a challenge [49]. Inappropriate interaction methods may reduce physical or psychological comfort [51], and thus lower the efficiency of human-robot collaboration [38], or even cause safety accidents [32]. Especially when people use AR technology to assist robots in authoring tasks like path adjusting [9, 10, 16, 22, 36] and visual programming [7, 23, 25, 40], the psychological discomfort caused by inappropriate paths [2] and the physical discomfort caused by inappropriate interactions become more prominent. For example, in specific task scenarios, people need to adjust the robot's pre-planned path to avoid collisions between the robot and human or object. If the interaction method causes discomfort, it may affect people's operation efficiency, experience, and physical safety.

In authoring task of AR-HRI, the interaction methods of existing researches mainly include gesture [36], head pose [36], gaze [48], touch screen [7], handheld device [35] and other unimodal interactions, as well as gesture + voice [36], head pose + voice [36], head pose + gesture + voice [36] and other multimodal interactions. These interaction methods are utilized to implement three operations: picking, moving and placing [36]. For example, for "head pose + voice", head point at target and say "pick" for picking target, and head move to move the target, and then say "place" for palcing the target. However, these studies have not fully considered whether the interaction methods used are in line with ergonomics in the current scenario, for example, different interaction methods may be suitable for different interaction areas [43], and different people may have different comfort feelings for different interaction methods [47]. Therefore, for this kind of AR-based authoring task, the existing interaction methods lack the interaction adjustment mechanism that adapts to the environment (general interaction area) and the human (individual comfort level), resulting in a serious reduction of comfort.

To solve the above problems, this paper proposes a dynamic dual-layer interaction adjustment mechanism (DDIA). Based on the principles of ergonomics, we invited participants to participate in experiments and established an **general layer model**, which divided the areas for appropriate interaction methods. Moreover, inspired by the "Dynamic Scene Adjustment (DSA)" proposed by Liu [31], we dynamically adjust the current interaction method based on the discomfort level (**individual layer model**), improving user comfort while ensuring interaction efficiency. Discomfort can cause stress [11], which in turn triggers a series of physiological responses, and the related physiological signals include Photoplethysmography (PPG) and galvanic skin response (GSR) [47]. Although unimodality could be weak and easily contaminated by noise like artifact [39, 45], combining the various modalities to overcome the weaknesses of each individual modality. Thus, we collected the participants' physiological signals (PPG and GSR) during the interaction process through experiments, and modeled and predicted the discomfort level by preprocessing and multimodal fusion of physiological signals [50]. Finally, we effectively combined the general layer and individual layer models to design a dynamic **dual-layer** interaction adjustment mechanism.

The contributions of this paper include:

1. **Dual-layer Interaction Adjustment Mechanism (DIAM)**. We proposed a DIAM for authoring task in AR-HRI, integrating general and individual factors. It uses an ergonomics-based general layer model and a physiological signal-based individual layer model to dynamically adjust AR interactions.

2. **Continuous Index Model**. We enhance discomfort prediction by shifting from a discrete to a continuous model in individual layers, adapting to individual differences, reducing errors, and aiding discomfort management.

3. **Validation Scenario**. This paper verifies the effectiveness and superiority of the dual-layer interaction adjustment mechanism that enhances user comfort and efficiency through path authoring and robot programming experiments.

## 2 RELATED WORKS

### 2.1 Ergonomics in Authoring of AR-HRI Task

The intersection of AR and HRI opens up new avenues for ergonomics research [43]. Previous studies [15] found that compared to pure physical HRI, users can better understand and optimize the ergonomics aspects of HRI by using AR to enhance Human-Robot Interaction (AR-HRI). This finding suggests that AR provides a valuable platform for assessing potential ergonomics issues such as interaction discomfort in Authoring of AR-HRI task.

Authoring of AR-HRI task mainly involves path adjusting and visual programming to achieve more efficient collaboration, avoid collision and conflict, adapt to dynamic environmental changes, and meet user's preferences and needs [6, 7]. The interaction methods involved in the current authoring of AR-HRI task include unimodal interaction (touch screen [7], handheld device [35], gesture [36], gaze [48], etc.) and multimodal interaction (gesture + voice [36], head pose + voice [36], head pose + gesture + voice [36], etc.). Compared to the mixed reality head-mounted display (HMD, such as Hololens 2), touch screen [7] and handheld device [35] may cause more ergonomics problems, such as: visual fatigue (constantly switching sight between the screen and the real environment), poor posture (users often need to lower their heads or bend over), limiting hand interaction, etc. [5]. Although wearing HMD for a long time may also cause head and neck discomfort [51], but since the path planning task is short [33], these problems can be alleviated by proper rest and adjustment [6]. In addition, HMD can also provide more natural and intuitive interaction methods (including gesture, gaze, voice, etc.), allowing users to fully immerse themselves in the task, rather than being limited by the device operation and interface. Given the close relationship between ergonomics and interaction comfort, there is a need for further maturation of ergonomics in the Authoring of AR-HRI task to enhance comfort during interaction.

### 2.2 Interaction Comfort Measurement

The measurement methods of interaction comfort include two main methods, namely subjective measurement methods (such as NASA TLX [21] and active reporting of physical discomfort [28], etc.) and objective measurement methods (such as GSR, PPG (converted to heart rate variability), skin temperature, electroencephalogram (EEG) and pupil measurement, etc.).

Subjective measurement. Since comfort is considered by most researchers as a subjective psychological response to environmental stimuli [12], subjective evaluation methods such as NASA TLX scale are usually considered as the most accurate human comfort

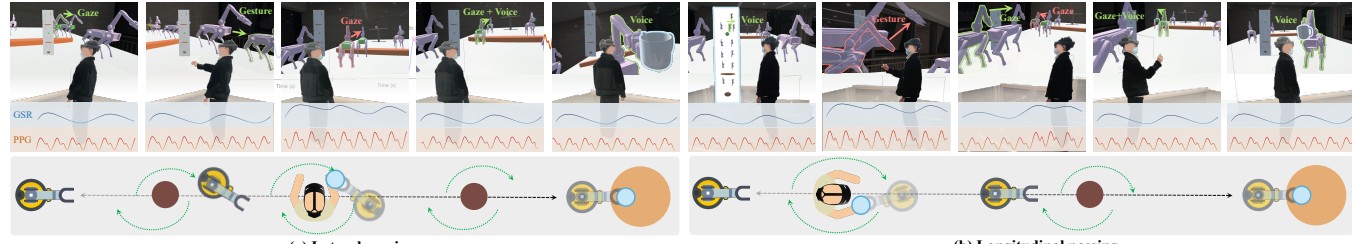

(a) Lateral passing    (b) Longitudinal passing

Figure 2: Task authoring of cup delivery with an initial path (grey) for data collection. The subject modifies the path (green) to avoid collisions, with two subtasks: (a) lateral and (b) longitudinal penetration. The robot starts from a direction (left or rear), goes around obstacles and the subject, gets the cup, hands it to the subject, and returns to the start.

measurement methods at present. But subjective scales cannot collect data in real time, and comfort data is highly discrete, sacrificing the accuracy of real comfort reflection [47].

Objective measurement. Physiological signals can objectively assess psychological and physiological responses, such as cognitive load, stress, and emotion etc. [29]. Specifically, the relationship between electrodermal activity and cognitive load, the relationship between heart rate variability and stress level, and the relationship between skin temperature and emotional state are the main evaluation indicators of interaction comfort [47, 51].

## 2.3 Learning from Physiological Signals

"Learning physiological signals" has appeared in various applications of understanding human comfort, including human-robot interaction [1, 4, 47] and augmented reality [42]. Physiological signal data is also used to implement various perception tasks, such as user experience quality [27], emotion recognition [46], attention assessment [14], the sense of co-presence. The latest flexible and non-invasive design of physiological signals creates new possibilities for understanding the human comfort that is invisible to the camera. Despite progress in using physiological signals to understand comfort, a gap exists in applying this to AR-HRI. The potential of dynamic interaction adjustment based on discomfort is underexplored. This leads to our research questions:

**RQ1**: How do GSR and PPG signals influence interaction comfort under different methods?

**RQ2**: What impact does the dual-layer interaction adjustment mechanism have on subjects' interaction experience?

These questions aim to address the research gap and enhance AR-HRI systems' comfort and efficiency.

## 3 METHOD

Our goal is to model and predict the interaction discomfort level. And the method of evaluating the discomfort level of interaction methods is to combine objective physiological signal measurement (GSR and PPG) and subjective scale (NASA TLX [21] items related to comfort and active reporting of physical discomfort [28]). Therefore, we devised an experiment for data collection and preprocessing.

**Participants and setup**. After obtaining the approval of the IRH, we invited 25 participants (aged between 21 and 29, of which 2 were female) to participate in the experiment. Among them, 2 had experience in using AR head-mounted devices (Hololens 2) before

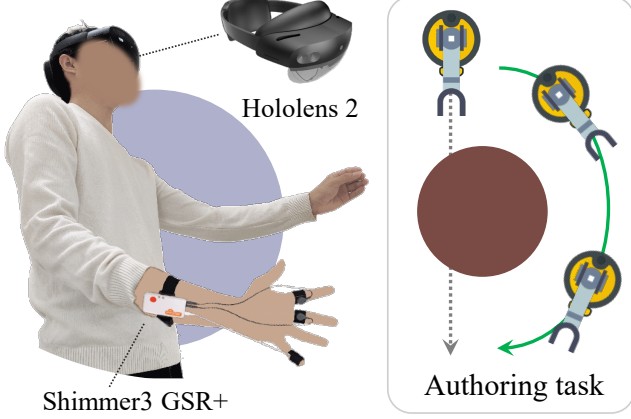

Figure 3: Device and data collection

the study. All participants were comfortable wearing the Hololens 2. Hololens 2 provides a resolution of 1440 × 936 for each eye, a refresh rate of 60 FPS, and a vertical/horizontal field of view (FoV) of 29°/43°. For each participant, we also attached a Shimmer3 GSR+ unit[1] [37] on the left hand (see Figure 3) to collect GSR and PPG data. In order to ensure that the GSR data monitoring is not affected by the arm and wrist movement, we asked the participants to interact with their right hand with the consent of the participants. We initially tested a wireless GSR wristband. Although the wristband is more portable and flexible compared to the finger sleeve (Shimmer3), it was excluded from the experiment due to its weak and unstable signal, as well as its low sampling rate. The Shimmer3 GSR+ unit has a frequency of 128Hz and transmits data to the PC with a delay of 25ms to 100ms.

**Interaction and authoring tasks**. During the study, each participant wore Hololens 2, kept their feet fixed in the scene, and completed the path authoring in the "cup delivery" task (see Figure 2) where the robot has planned the initial path (gray). The cup delivery task involves the robot moving from its starting position to the table (brown), picking up the cup (blue circle), handing it to the human, and then returning to the original position upon receiving human instructions. However, the initial path goes through the subject and obstacles. To ensure safety and no collision, the subject

---

[1]https://www.shimmersensing.com/

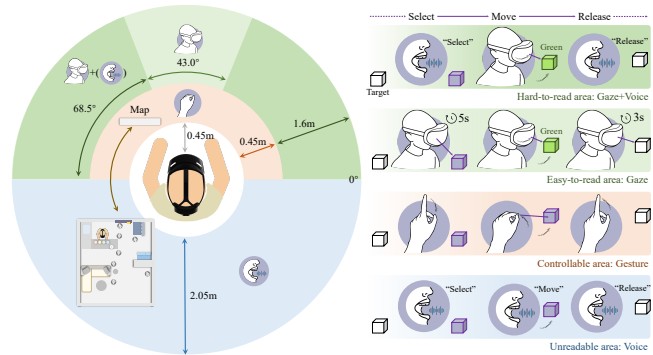

**Figure 4: Modeling of general layer interaction areas.**

needs to adjust the initial path with the picking, moving and placing operations of the target object by gesture (grab), gaze, voice, voice + gaze (see Figure 4) to get the adjusted path (green). Figure 2 shows the 2 sessions of Authoring tasks (lateral and longitudinal passing), and the sessions settings for each participant are the same, each session lasts about 10 minutes, and each session needs to be done 4 times, allowing users to actively report physical discomfort during the experiment. In order to alleviate the head discomfort caused by wearing Hololens 2 for too long and avoid posture drift, the participants were allowed to rest appropriately during the experiment. Upon session completion, participants are required to sequentially evaluate their interactions using two elements of the NASA TLX subjective scale: Physical Demand and Frustration Level. This process typically takes approximately 3 minutes. The items in the scale have 1 (very low) to 5 (very high) rating points to choose from, and the higher the score, the more discomfortable.

**Duration of the study**. Including pre-questionnaires (demographics, AR-HRI-related experience questions), software and hardware settings (Hololens visual calibration, physiological signal transceiver adapter deployment), pre-study guidance, warm-up training, task sessions (completion of tasks, monitoring of physiological signals, self-evaluation of interaction methods), rest and debriefing, each participant's study took about 1 hour. In total, we collected about 2 hours of GSR and PPG data paired with comfort and time-synchronized. We only retained the data with obvious fluctuations in GSR and PPG signals, and removed the data with insignificant fluctuations or abnormal jumps. We retained approximately 1 hours of GSR and PPG data.

**Data processing and analysis**. The original physiological signals cannot be used directly, because they exhibit: 1) non-stationary behavior: the statistical characteristics of the signals change over time, which makes the signal processing complex; 2) complex characteristics: the features extracted from GSR signals are very complex, and their detailed evaluation is insufficient; 3) transient artifacts: the signals may contain fast transient artifacts, which need to be eliminated by applying a noise reduction filter with post-processing tools; 4) individual differences: the same stimulus may cause different intensity of physiological responses in different people. Since physiological signals are sensitive to motion artifacts, we instructed participants to act naturally, but also to prevent them from waving their hands excessively [17]. Therefore, we performed

a series of preprocessing on the original signals, including segmentation, resampling, filtering, etc. (see Figure 5(a)):

1) **Segmentation operation**: The original GSR and PPG signal was cut into 7481 samples, each sample window size was 5s, and the window moving step size was 0.25s.

2) **Resampling**: In order to solve the long tail effect caused by individual differences, the sample labels of the data set were adjusted from five (1, 2 and 3 means normal, 4 and 5 means discomfortable) to two groups (sample size: discomfortable (group D) and normal (group N) each have 3200).

3) **Filtering**: In order to reduce the impact of noise and improve the quality of the signal, third-order Butterworth filter was used to preprocess each PPG time segment, and median smoothing filter was used to preprocess each GSR time segment, and decomposed it into SCL and SCR components [50].

**Result and discussion**. We assess sample quality by analyzing differences between group D and N, and among different interactions. The reasons we do not perform the analysis of variance (ANOVA) on continuous time series signals, but on the wavelet features of the signals[26] is: 1) **Dimensionality reduction**: Wavelet features can greatly reduce the dimensionality of data, making data processing and analysis more efficient. 2) **Information extraction**: Wavelet features can effectively capture key information in GSR signals. 3) **Stability**: Wavelet features are more stable and less susceptible to noise and outliers, especially the wavelet function db4 has the best denoising effect. 4) **Interpretability**: Wavelet features have good interpretability, which helps us understand the relationship between GSR and interaction comfort. Since the data of PPG and GSR do not pass the normality test and homogeneity of variance test, we use one-way ANOVA (non-parametric method, Kruskal-Walls) to test for significant differences. Group D and N have significant differences ($p < .001$) in all wavelet features (variance, mean, maximum, minimum, energy) of GSR and PPG. The lower values of the group N of wavelet features indicate that the comfortable participants have less variation, lower average level, lower peak, lower minimum and less energy in their GSR and PPG signals. This may mean that they are less aroused and emotional than the discomfortable participants. This is consistent with previous studies that found that GSR and PPG signals are sensitive to discomfort [47]. The ANOVA results suggest that GSR and PPG signals' wavelet features are sensitive to AR-HRI interaction discomfort. Consequently, these signals could be effectively used to model and predict discomfort in AR-enhanced HRI scenarios. However, no significant differences exist in the wavelet features of physiological signals (GSR and PPG) across the four interaction methods. Therefore, while these signals can predict an individual's comfort with their current interaction method, they are unable to differentiate among various interaction methods, nor can they determine the subsequent interaction approach based on the current one.

The timing of adaptively switching interaction methods based on discomfort depends not only on the comfort of the individual's current interaction method but also on the general applicability of different interaction methods. Therefore, we have designed a dynamic dual-layer interaction adjustment mechanism including general layer modal and individual layer modal.

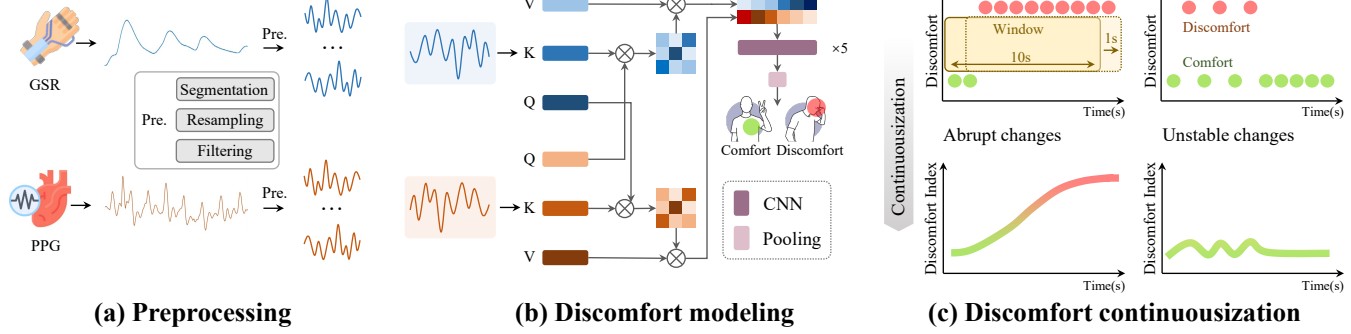

**Figure 5: Modeling Continuous Discomfort Index in individual layer discomfort.**

## 3.1 General Layer Modeling

**General Layer Interaction Adjustment Mechanism**. By recording and analyzing their operation effects and feedback, we obtained the applicability of each interaction method under different conditions. Ultimately, the general layer interaction area modeling is constructed by statistically analyzing the average applicable area of the normal distribution of each interaction method. Figure 4 details the most suitable interaction methods (automatic switching), the distance range and viewing angle range of the interaction methods for different areas: **Controllable area (red)**: The distance range of this area is from 0.45m to 0.90m, the interactive angle range is from 0° to 180°, and the recommended interaction method is gesture (grab). **Readable area (green)**: The distance range of this area is from 0.90m to 2.50m, the interactive angle range is from 0° to 180°, the recommended interaction method is gaze (for the easy-to-read area), and the alternative interaction methods are voice + gaze (for the hard-to-read area, dark green). **Unreadable area (blue)**: The distance range of this area is from 0.45m to 2.50m, the interactive angle range is from 180° to 360°, and the best interaction methods are voice. Hololens 2 screen left side provide a map with path points and different areas for voice interaction in this area (see Figure 4).

For the final placement position of the target object, we refer to the three interaction categories proposed by Hall [20]: intimate (0-0.45 m), personal (0.45-1.20 m) and social (1.20-3.60 m). Considering the physical safety in the subsequent human-robot interaction tasks, we recommend users to place the target interaction object outside the intimate distance. This can ensure that both users and robots can operate within a safe distance during human-robot interaction, while also ensuring the efficiency and effect of interaction.

However, this general layer model may have bias, only considering the average, without considering the individual differences of each participant, for example, some participants (P1, P4-7, P10, P11, P13, P17-18, P20) felt gesture interaction discomfort before reaching the best gesture interaction area boundary divided by the general layer model, then we need to further investigate the interaction that this type of participant most wants to switch to at this time.

In order to reduce the impact of individual differences on the interaction comfort in the general layer model, we collected and analyzed the participants' physiological signals (PPG and GSR) to establish a individual layer comfort model. This model can predict the user's discomfort level when performing different interaction

methods, and provide personalized interaction method recommendations. By combining the general layer interaction area modeling and the individual layer comfort model, we finally constructed a dual-layer interaction adjustment mechanism. This mechanism can dynamically select the most suitable interaction method according to the user's actual situation and environmental conditions, thereby improving the interaction efficiency and user experience.

## 3.2 Individual Layer Modaling

We built a computational model by analyzing the GSR and PPG data paired with comfort. Although subjective scales can measure comfort, they may interfere with normal activities and be affected by recall bias and social expectations, reducing the reliability and validity [41]. Therefore, we propose an interaction discomfort prediction model that combines physiological signals and subjective evaluation, and use an open function to describe the change of GSR and PPG driven discomfort. We also use the collected data to fit these functions with deep learning models, so as to predict interaction discomfort level.

**Cross-attention discomfort prediction**. Before model training, we preprocess the collected raw data (see data processing and analysis in 3.2 for details). The preprocessed data was randomly split into training and validation sets, each with 3200 samples, with a ratio of 8:2, and a random seed of 5000 was designed. Our inputs are preprocessed physiological time series signals (time segments), and the outputs are discomfort or normal. Supervised learning has proven that by learning physiological features, it can quickly, accurately, and robustly detect cognitive states [34]. Thus, compared to manually extracting features from the processed GSR signal as sample attributes, we use deep networks (5 layers of CNN and 1 layer of pooling) to automatically learn the features of time series signals, avoiding human bias, improving feature quality and generalization. We can choose the feature extraction method according to the characteristics of the time series signal, such as CNN extracting the short-term local dependency pattern of the PPG and GSR signal, and can map the time series signal to the same dimensional feature space, retaining more information, which is convenient for multimodal fusion[50]. Inspired by [50], we use the Bidirectional Cross Attention module (see Figure 5(b)) to address the delay issue between multimodal physiological signals and align and fuse PPG and GSR signals more effectively with cross-attention mechanism:

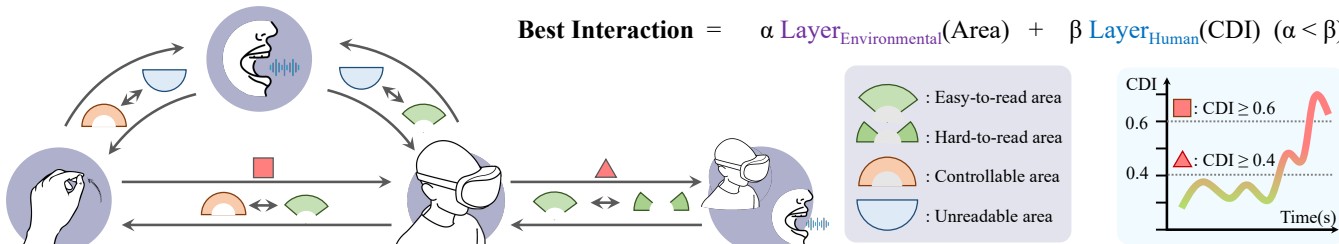

Figure 6: Individual-layer interaction adjustment mechanism.

$$\text{Attention}(x, y_i) = \sum_{i=1}^{N} \text{softmax}\left(\frac{xW_Q(y_iW_K)^T}{\sqrt{L}}\right)y_iW_V, \quad (1)$$

where $x$ is the query, $y_i$ is the value, $W_Q$, $W_K$ and $W_V$ are parameter matrices, $\alpha_i$ is the attention weight, indicating the matching degree of $x$ and $y_iW_K$. The query (Q), key (K) and value (V) are mapped to the dimension $L$. PPG and GSR are used as queries, keys and values interchangeably, and cross attention weights are calculated to obtain PPG and GSR representations fused with each other's information.

Figure 5(b) shows discomfort modeling and prediction. After 916 epochs, accuracy of our continuous signals (0.928 ± 0.012) is higher than discrete features (0.794 ± 0.051) [50], possibly due to detail loss or noise in discrete features, while continuous signals preserve more temporal information and signal changes. Considering device delay of 25ms to 100ms, one layer of cross-attention ensures at least one update per second, while more layers cannot.

**Enhancing discomfort prediction from discrete to continuous** can more accurately reflect the changes in the discomfort level of the subjects, rather than categorizing them into discrete classes [19]. This method can better adapt to individual differences and dynamic changes, alleviate misjudgments of the subjects' discomfort, and thus better assist the subjects in managing discomfort. Specifically, windows ($w$, monitored every second with a 10s window and 1s step, see Figure 5(c)) that are entirely within the discomfort and normal intervals are assigned values of 1 and 0, respectively. However, if a window involves two states, its Continuous Discomfort Index (CDI(x)) is calculated using the following formula:

$$CDI(x_n) = \begin{cases} e^{-\frac{(x_n - m)^2}{2\phi^2}} & \text{if } t_{\text{normal}} < w < t_{\text{discomfort}} \\ 1 - e^{-\frac{(x_n - m)^2}{2\phi^2}} & \text{if } t_{\text{discomfort}} < w < t_{\text{normal}} \end{cases} \quad (2)$$

where $m = 0$, $\phi = 0.3$, $x_n = -1 + 0.1 \times n$, $n = 1, \ldots, 10$ are used to generate a uniformly distributed sequence from -1 to 0, representing the position of the time window. Then, the discomfort within these 10 seconds is gradually mapped to the range of 0 to 1 using a normal distribution (see Figure 5(c)). The purpose of this is to assign a fuzzy discomfort index to each time window, rather than a discrete discomfort category. By transforming discrete predictions into a continuous form, we effectively address the issues of abrupt changes and unstable changes (see Figure 5(c)). This approach allows us to more accurately reflect the actual feelings of the subjects, rather than simply categorizing them as "normal" or "uncomfortable". Therefore, this method can more effectively help users manage discomfort. Finally, the CDIs are sent to Hololens via UDP (User Datagram Protocol), enabling interaction adjustments as needed.

**Individual Layer Interaction Adjustment Mechanism**. To address the "bias" issue (limited applicability, i.e., the interactive adjustment strategy cannot satisfy individual users.) inherent in the general layer model, we once again invited 25 participants to engage in authoring tasks. We found that more than half of the participants reported discomfort in both the "controllable area" and the "difficult-to-read area". Specifically, for the "controllable area", 64% of participants (all with discomfort indices exceeding 0.6) reported discomfort and requested a switch to gaze interaction. For the "difficult-to-read area", 72% of participants (all with discomfort indices exceeding 0.4) reported discomfort and requested a switch to gaze+voice interaction. Figure 6 illustrates the individual layer interaction adjustment mechanism. When a discomfort index exceeding 0.6 is detected, the system automatically switches to gaze interaction if the participant is in the "controllable area". If the participant is in the "difficult-to-read area", the system automatically switches to gaze+voice interaction when a discomfort index exceeding 0.4 is detected. This mechanism allows participants to choose the interaction method that best suits their comfort and preferences. Therefore, the priority of the individual layer interaction adjustment mechanism is higher than that of the general layer interaction adjustment mechanism.

## 4 EVALUATION

We evaluate the performance of our dual-layer interaction adjustment mechanism in dynamically adjusting interactions to reduce discomfort through a series of objective measurements. We demonstrate the effectiveness and superiority of the dual-layer interaction adjustment mechanism considering both general and individual dimensions through experiments with ecological validity [30].

## 4.1 Authoring of Real-life AR-HRI Tasks

**Participants and setup**. To mitigate carryover effects [8], we recruited 31 distinct participants (aged 20-29, including 3 females), separate from the 25 discussed in Section 3.1. Two of them had experience using AR headsets before the study. Our study design is mixed-model, with the interaction strategies varying as a between-subjects factor. Within-subject factors encompass two task difficulty levels. The easy level involves completing all path adjusting tasks before all robot programming tasks (separate tasks). The difficult

level requires alternating between path adjusting and robot programming after each task (combine tasks). All participants reported that their device-wearing conditions were comfort, and the hardware and software setup is similar to Section 3.2 (see Figure 3). All participants were evenly divided into three groups, Group A (10 males, 1 female, M=23.4, SD=2.77) chose the interaction method (**baseline**) according to their wishes, Group B (9 males, 1 female, M=24.0, SD=2.40) used the general layer interaction switching mechanism (**single layer** only), and Group C (9 males, 1 female, M=23.4, SD=2.50) used the **dual-layer** interaction adjustment mechanism. Each condition required the completion of two types authoring tasks: path adjusting and programming (see Figure 2).

**Real-life authoring tasks**. Figure 1 illustrates two tasks for "Welcoming Guests": path authoring (Task 1) and robot programming (Task 2). Participants, wearing Hololens 2 and stationed in the kitchen scene, engaged in two activities. In path authoring, they modified the robot's initial path (gray) to accommodate four tasks: door opening, cup grabbing, obstacle avoidance, and cup placing. This was achieved by programming the robot's behaviors using the interaction methods shown in Figure 4. During robot programming, participants assigned tasks to the robot by moving colored balls (red, yellow, green, and blue) representing the four tasks to the task setting area beside the corresponding robot's virtual avatar. For instance, the "door opening task ball" was placed in the task setting area of the virtual robot near the door.

**Task duration**. The two authoring tasks need to be conducted in two sessions with two different difficulties. A Latin square design was employed to balance the sessions. Each round takes approximately 10 minutes, with path authoring and robot programming each requiring about 5 minutes. The study for each participant also takes about 45 minutes including pre-questionnaires, software and hardware settings, pre-study guidance, warm-up training, task sessions, after-task subjective scales, rest and debriefing. Each session was monitored to ensure that the subjects' movements were within a safe range.

## 4.2 Result

We evaluate the effectiveness and superiority of the dual-layer interaction adjustment mechanism through a combination of subjective and objective measures. Objective behavioral data reflect the operation efficiency and operation load of users under different interaction methods. We collected the average interaction time per session, and the total number of operations for each task . We also constructed scales using 7-point Likert-style items to measure participants' subjective feelings and preferences, with higher scores being more positive. These scales rated physical and psychological comfort (2 items, Cronbach's $\alpha$ = .758), acceptance of interaction adjustment strategies (2 items, Cronbach's $\alpha$ = .731), and overall design usability (2 items, Cronbach's $\alpha$ = .745). All collected data have undergone normality testing (Shapiro-Wilk) and homogeneity of variance testing (Levene's). Meanwhile, we compared each participant's predicted discomfort with their actual experience, assessing the alignment between predicted and perceived uncomfortable interactions. Our model's discomfort predictions are statistically significant, with an 75.2% accuracy (see Figure 7(a)), indicating a minimal chance of them being random.

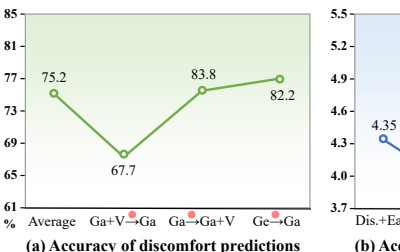
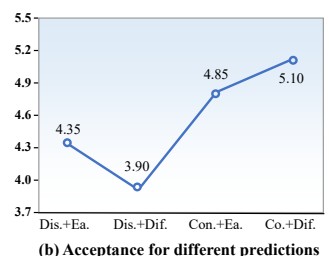

(a) Accuracy of discomfort predictions          (b) Acceptance for different predictions

**Figure 7: Evaluation of discomfort prediction accuracy and interaction strategy acceptance**

**Between-subject analysis.** We used ANOVA (Welch's method) to determine whether there were significant differences among the three groups (see Figure 1). **Single-session interaction duration**: C group (147.2 seconds) and B group (215.8 seconds) had significantly shorter interaction time than A group (270.0 seconds), $F_{(2, 37.7)}$ = 552.67, $p < .001$. Games-Howell post hoc test showed significant differences among three groups ($p < .001$). **Total number of operations in robot programming**: C group (10.8 times) and B group (13.5 times) had significantly fewer number of operations than A group (21.4 times), $F_{(2, 37.7)}$ = 134.4, $p < .001$. Games-Howell post hoc test showed significant difference between A and B group ($p < .001$) and C group ($p < .001$). **Total number of operations in path adjusting**: C group (13.7 times) and B group (19.4 times) had significantly fewer number of operations than A group (23.9 times), $F_{(2, 37.7)}$ = 88.40, $p < .001$. Games-Howell post hoc test showed significant differences among three groups ($p < .001$). **Comfort level during interaction**: C group (4.60 points) had significantly higher comfort level than B group (3.73 points) and A group (2.48 points), $F_{(2, 38.6)}$ = 52.1, $p < .001$. Games-Howell post hoc test showed significant difference among three groups ($p < .005$). **Acceptance level of interaction adjustment strategy**: C group (4.97 points) and B group (4.35 points) had significantly higher acceptance level than A group (3.02 points), $F_{(2, 38.7)}$ = 66.0, $p < .001$. Games-Howell post hoc test showed significant differences between A and B ($p < .001$) and C group ($p < .001$), B and C ($p = .007$). **Usability**: C group (4.88 points) and B group (4.38 points) had significantly higher Usability than A group (2.61 points), $F_{(2, 38.8)}$ = 86.8, $p < .001$. Games-Howell post hoc test showed significant differences between A group and B and C group ($p < .001$).

**Within-subject analysis.** We used ANOVA to investigate the main effect of task difficulties and their two-way interaction effects with interaction adjustment strategies. Participants in easy level task had a significantly shorter interaction time (197.1s) compared to those in difficult level task (228.7s), $F_{(1, 60)}$ =106.99, $p < .001$. In the robot programming task, participants in easy level performed significantly fewer operations (13.9 times) than those in difficult level (16.9 times), $F_{(1, 60)}$ = 28.7, $p < .001$. However, in the path adjusting task, there was no significant difference between participants in easy (18.2 times) and difficult level (20.0 times). The only two-way interaction effect we found was in the operations of the robot programming task, $F_{(2, 37.7)}$ = 11.2, $p < .001$.

**Discrete vs. Continuous Discomfort Prediction**. In addition to continuous discomfort prediction, participants in Group C were

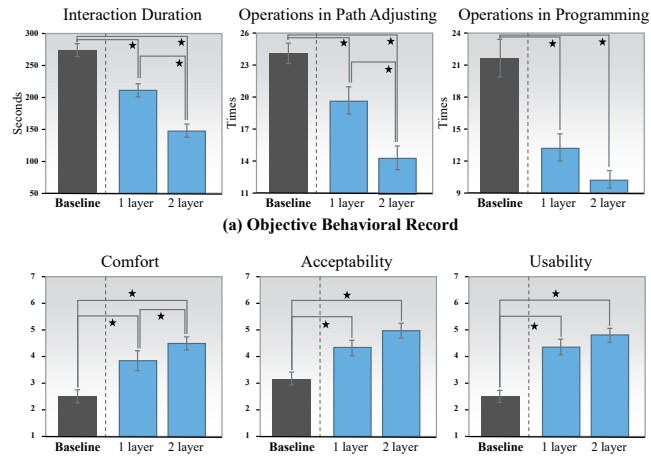

(a) Objective Behavioral Record

(b) Subjective Evaluation

**Figure 8: An ANOVA (Welch's method) compared subjective and behavioral measures among three groups in the validation study. Statistical significance is marked with stars.**

asked to complete two sessions involving discrete discomfort prediction and evaluate their acceptance of interaction adjustment strategies. A significant difference was observed between discrete (4.13 point) and continuous (4.97 point) discomfort prediction, F(1, 38)=22.6, p < .001 (see Figure 7(b)).

### 4.3 Discussion

This study aims to compare the impact of three different interaction adjustment strategies on user performance and experience in two authoring tasks of AR-HRI. The results show that Group C, which uses a dynamic dual-layer interaction adjustment mechanism (DDIA), and Group B, which uses an single-layer interaction adjustment mechanism, significantly outperform Group A, where users choose the interaction method according to their wishes, in terms of single session interaction duration, total number of operations, comfort during interaction, acceptance of interaction adjustment strategies, and usability. This suggests that by providing more flexible and efficient interaction methods, user efficiency and satisfaction in authoring tasks of AR-HRI can be improved. In addition, the results also show that Group C, which uses a dual-layer interaction adjustment mechanism, significantly outperforms Group A and B in terms of the interaction duration, total number of operations in path adjusting, and comfort level during interaction. This indicates that compared to the single-layer interaction adjustment mechanism, the DDIA can better adapt to individual differences and dynamic changes, reduce misjudgments, and effectively help participants manage discomfort. The results also revealed that separating tasks (easy) improved efficiency in the robot programming task but showed similar performance to combining tasks (difficult) in the path adjusting task, indicating its effectiveness may be task-dependent.

The two-way interaction effect was only seen in the robot programming task, likely due to its complex and decision-based nature, which might be affected by discomfort prediction and adjustment

methods. The path adjusting task, which is more procedural, didn't show this effect. Despite both tasks being part of the AR-HRI authoring task, they demand different cognitive skills: strategic planning for robot programming and spatial reasoning for path adjusting.

Moreover, compared to discrete discomfort prediction, continuous prediction could enhance the acceptance of interaction adjustment strategies. It suggests that a continuous model could adapt to dynamic changes, minimize misjudgment, and aid discomfort management.

## 5 LIMITATIONS AND FUTURE RESEARCH

This study considers the impact of GSR and PPG on interaction comfort. However, due to the need for per-second updates and computational resource overhead for discomfort monitoring, it does not take into account additional physiological signals such as EEG [41, 51]. This may result in our model missing some crucial information. Furthermore, this work only considers the physiological state of users during interaction with robots, without taking into account factors such as personality traits and emotional states, which could also influence user interaction comfort [46].

Moreover, the development process necessitated certain constraints on AR usage, such as limiting interactions to the right hand to accommodate sensor placement on the other hand. This constraint naturally restricts use-cases, particularly bi-manual tasks. To overcome this, we aim to use head-mounted devices for signal collection in future iterations.

Future research plans include incorporating other types of physiological information and psychological states to more comprehensively assess user comfort in VR/AR. On the other hand, we can leverage probabilistic modeling and learning techniques to extract more statistical variations and individual differences from physiological signals, enhancing the generalizability and robustness of the model. Moreover, recommendations for switching to more comfortable interaction methods should consider the impact of specific scenario factors. For instance, users engaged in cooking with both hands occupied should be advised against interaction methods related to gestures. Additionally, future work needs to consider the differences in physiological data collected under different interaction methods [18].

## 6 CONCLUSION

This paper proposes a dynamic dual-layer interaction adjustment mechanism (DDIA) for authoring of AR-enhanced Human-Robot Interaction (AR-HRI) tasks. The mechanism consists of an general layer model and a individual layer model, which can dynamically switch to a appropriate interaction method. The general layer model divides the interaction areas according to the principles of ergonomics, and the individual layer model predicts the user's discomfort level, dynamically adjusts interaction methods based on physiological signals. We evaluate the performance of our mechanism through objective and subjective measures. The results show that our DDIA mechanism can significantly improve the user's comfort, and efficiency in robot authoring tasks. Our work contributes to the ergonomics research of AR-HRI, and provides a valuable platform for assessing and improving the interaction comfort in authoring of AR-HRI tasks.

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
