# OpenReview forum: "Improving Interaction Comfort in Authoring Task in AR-HRI through Dynamic Dual-Layer Interaction Adjustment"
_acmmm.org/ACMMM/2024/Conference — MM2024 Oral_

### Official Review · Reviewer_UZQr · 2024-05-17

**Rating:** 4
**Confidence:** 3

**Summary:**

This paper proposes a method to improve the interactive comfort of task creation in augmented reality human-computer interaction by dynamic double-layer interaction adjustment. This method reduces misjudgment and improves comfort management by dynamically adjusting the interface. The author verifies the success of the mechanism in task creation, proves its effectiveness, significantly promotes augmented reality human-computer interaction, and promotes more comfortable and efficient human-centered interaction, which is somewhat innovative. This paper proposes a method to improve the interactive comfort of task creation in augmented reality human-computer interaction by dynamic double-layer interaction adjustment. This method reduces misjudgment and improves comfort management by dynamically adjusting the interface. The author verifies the success of the mechanism in task creation, proves its effectiveness, significantly promotes augmented reality human-computer interaction, and promotes more comfortable and efficient human-centered interaction, which is somewhat innovative.

**Strengths:**

This method reduces misjudgment and improves comfort management by dynamically adjusting the interface. The author verifies the success of the mechanism in task creation, proves its effectiveness, significantly promotes augmented reality human-computer interaction, and promotes more comfortable and efficient human-centered interaction, which is somewhat innovative. This paper proposes a method to improve the interactive comfort of task creation in augmented reality human-computer interaction by dynamic double-layer interaction adjustment. This method reduces misjudgment and improves comfort management by dynamically adjusting the interface. The author verifies the success of the mechanism in task creation, proves its effectiveness, significantly promotes augmented reality human-computer interaction, and promotes more comfortable and efficient human-centered interaction, which is somewhat innovative.

**Limitations:**

1. In the third part of the method, the age and gender of 25 participants are explained, but the height and weight of these participants are not mentioned. Since the objective physiological signals (GSR and PPG) to be measured are related to height and weight, it is recommended to add clarification;
2. The task process in Figure 2 is relatively simple, so it is suggested to draw it in more detail;
3. In Section 3.2, the description of the cross-attention mechanism fusing each other's PPG and GSR information is not detailed enough;
4. Chart annotation: the annotation of some charts is relatively simple, lacking detailed chart description and explanation;
5. Due to the lack of effective comparative tests in section 4, other experiments using AR technology to assist robots to complete path adjustment should be added for comparison.

**Suitability:**

2

---

### Official Review · Reviewer_bxee · 2024-05-22

**Rating:** 6
**Confidence:** 1

**Summary:**

The paper presents a novel approach to enhancing user comfort and efficiency in augmented reality-enhanced human-robot interaction (AR-HRI) tasks. It introduces a dynamic dual-layer interaction adjustment mechanism (DIAM), comprising a general layer model based on ergonomic principles and an individual layer model using physiological signals (PPG and GSR) to predict and respond to user discomfort. The study involved 25 participants using AR headsets and sensors to perform robot path adjustment and programming tasks. The results showed that DIAM significantly improved interaction efficiency and comfort compared to traditional methods.

**Strengths:**

1. The innovation of this article is the introduction of the Dual Layer Interaction Adjustment Mechanism (DIAM). This mechanism dynamically combines ergonomic principles with real-time physiological feedback to provide a solution for enhancing user comfort in AR-HRI tasks.
2. I don't have enough background knowledge about the technical implementation of DIAM, but from the article it seems to be methodologically sound. The authors use deep learning techniques to process and predict the discomfort of physiological signals, and the preprocessing steps for physiological data (segmentation, resampling, filtering) seem reasonable (but I'm not an expert, and need to refer to other reviewers).
3. This paper presents an evaluation of the proposed mechanism through a carefully designed experiment involving 25 participants. The experiment was tightly controlled and participants performed realistic tasks using AR headsets and sensors. The study evaluated objective metrics (interaction time, number of operations) and subjective metrics (comfort, acceptance) to provide a comprehensive assessment of the effectiveness of DIAM.
4. well-organized and clearly written, with detailed descriptions of the methodology, experiments, and results. Figures and diagrams effectively illustrate the concepts and findings, making it easier for readers to understand.

**Limitations:**

I like the idea of this article and I don't think it has very obvious limitations. And I do agree with the limitations that the author proposes in section 5. I think that the physiological signal collection and interaction methods need to be adapted accordingly when faced with different interaction tasks. The authors suggest that in the future, a head-mounted device can be used for signal collection, and I am concerned whether the head-mounted device will conflict with the head-mounted AR headset and affect the user's comfort experience? And in my experience head-mounted devices may not always collect accurate data, and are easily affected by hair. Anyway, I think this article is a good research article.

**Suitability:**

2

---

### Official Review · Reviewer_XZCb · 2024-05-24

**Rating:** 6
**Confidence:** 3

**Summary:**

The paper explores the integration of physiological signals and interaction comfort in augmented reality (AR) for human-robot interaction (HRI). The study introduces a dynamic dual-layer interaction adjustment mechanism (DDIA) to enhance user comfort and interaction efficiency.

In their research, the authors delve into the impact of physiological signals, such as photoplethysmography (PPG) and galvanic skin response (GSR), on interaction discomfort. By continuously monitoring these signals, the study aims to predict and manage discomfort levels, thereby adapting interaction methods dynamically. This approach helps in mitigating the stress and discomfort associated with prolonged or awkward interaction methods, ensuring a more comfortable user experience.

The authors validate the effectiveness of the dual-layer modeling through extensive experiments. Their results demonstrate that the DDIA mechanism significantly outperforms traditional single-layer models and user-selected methods in terms of reducing interaction discomfort and enhancing operational efficiency. This innovative approach highlights the potential for more ergonomic and user-friendly AR-HRI systems, paving the way for future advancements in this field.

The study's findings contribute valuable insights into the importance of physiological signal integration for interaction comfort and provide a robust framework for future research and development in AR-HRI tasks.

**Strengths:**

This paper presents an outstanding exploration into the integration of physiological signals to predict discomfort in AR-HRI interactions. The idea of using physiological signals to manage interaction comfort is innovative and impressive. However, given the unstable nature of these signals, which are easily influenced by human movements, implementing this idea is challenging. The authors have done an excellent job in the data analysis section, meticulously working through various methods until they found an effective way to correlate physiological signals with discomfort.

Moreover, the concept of employing both a general layer and an individual layer is commendable. This dual-layer approach ensures that the interaction design is not starting from scratch but is built on a solid foundation. Simultaneously, it allows for personalization to accommodate individual differences and temporal variations. This idea is not only applicable to the specific context of the paper but also holds significant potential for other fields.

The evaluation section is particularly convincing. The results show many significant p-values (p < 0.001) across multiple comparisons, reinforcing the robustness of the findings. Importantly, these results are based on real-life AR-HRI tasks rather than simplified experimental scenarios, which adds substantial credibility to the conclusions.

The paper is well-written, concise, and informative, making it a pleasure to read. The figures and illustrations are beautifully crafted, enhancing the overall presentation and comprehension of the content. Overall, I highly recommend this paper for publication and believe it will make a valuable contribution to the field.

**Suitability:**

3

---

### Official Review · Reviewer_99ai · 2024-06-03

**Rating:** 5
**Confidence:** 2

**Summary:**

The paper, titled "Improving Interaction Comfort in Authoring Task in AR-HRI through Dynamic Dual-Layer Interaction Adjustment," addresses the challenge of selecting interaction methods to enhance physical comfort in Augmented Reality Human-Robot Interaction (AR-HRI) scenarios. It proposes a dynamic dual-layer interaction adjustment mechanism that includes a general layer model, based on ergonomics, and an individual layer model that uses physiological signals to predict user discomfort. This mechanism dynamically adjusts interactions to improve user comfort and interaction efficiency, validated through authoring tasks like path adjusting and programming in AR-HRI.

**Strengths:**

The paper provides a comprehensive summary of common interaction areas including distances, and ranges, along with corresponding interaction methods, contributing to a better understanding of effective interaction design in AR-HRI. Another strength is the integration of both Galvanic Skin Response (GSR) and Photoplethysmogram (PPG) as objective measurements within a deep learning model to predict individual discomfort probabilities. This dual-layer approach, incorporating both ergonomic principles and physiological signals, is innovative and enhances the adaptability and personalization of the interaction methods. Furthermore, the paper employs a robust evaluation methodology, using subjective evaluations and objective behavioral records to measure the usability and acceptability of the dual-layer system. Experimental results demonstrate significant improvements in user’s comfort and interaction efficiency compared to single-layer and baseline methods. The clarity and detail in presenting the experimental setup and results further enhance the paper's impact and applicability.

**Limitations:**

The evaluation, though thorough, is limited to specific authoring tasks in controlled environments, which may not fully capture the variability of real-world AR-HRI applications. Additionally, the paper could benefit from comparisons with other state-of-the-art methods to provide a clearer benchmark of its relative performance and potential limitations in broader contexts.

**Suitability:**

3

---

### Meta-Review · Area_Chair_cc8n · 2024-07-02

**Recommendation:** Accept (Oral)
**Confidence:** 4

**Metareview:**

The paper "Improving Interaction Comfort in Authoring Task in AR-HRI through Dynamic Dual-Layer Interaction Adjustment" proposes a novel approach to enhance physical comfort in Augmented Reality Human-Robot Interaction (AR-HRI). The study introduces a dynamic dual-layer interaction adjustment mechanism that combines ergonomic principles with physiological signals to predict and manage user discomfort. The effectiveness of the mechanism is validated through experiments involving AR-HRI tasks such as path adjustment and programming.

The paper has received unanimous feedback from the reviewers, appreciating the innovative use of physiological signals and the robust evaluation methodology.

Based on the reviews, the paper is recommended for acceptance. The paper presents a significant contribution to AR-HRI by proposing an innovative and well-validated approach to enhancing user comfort and interaction efficiency. While there are some limitations, such as the controlled evaluation environment and lack of detailed comparisons, the strengths and practical applicability of the work justify its acceptance. The authors are encouraged to address the feedback provided by reviewers, as detailed in the rebuttal.